# Critical Analysis of the Value of Drought Information and Impacts on Land Management and Public Health

**Tingting Liu** [1,*] **, Kelly Helm Smith** [1] **, Richard Krop** [2] **, Tonya Haigh** [1] **and Mark Svoboda** [1]

1   National Drought Mitigation Center, University of Nebraska-Lincoln, 3310 Holdrege Street, 821 Hardin Hall, Lincoln, NE 68583-0988, USA; Ksmith2@unl.edu (K.H.S.); thaigh2@unl.edu (T.H.); msvoboda2@unl.edu (M.S.)
2   The Cadmus Group, LLC, 1620 Broadway, Santa Monica, CA 90404, USA; Richard.Krop@cadmusgroup.com
*   Correspondence: tliu19@unl.edu

**Abstract:** This paper reviews previous efforts to assign monetary value to climatic or meteorological information, such as public information on drought, climate, early warning systems, and weather forecast information. Methods and tools that have been explored to examine the benefits of climatic and meteorological information include the avoided cost, contingent valuation, choice experiments, benefit transfer, and descriptive approaches using surveys. The second part of this paper discusses specific considerations related to valuing drought information for public health and the Bureau of Land Management. We found a multitude of connections between drought and the land management and health sectors in the literature. The majority of the papers that we summarized only report biophysical change, because the economic losses of drought are not available. Only a few papers reported economic loss associated with drought. To determine the value of drought information, we need to know more about the role it plays in decision making and what sources of drought information are used in different sectors. This inventory of methods and impacts highlights opportunities for further research in valuing drought information in land management and public health.

**Keywords:** drought; climatic information; land management; public health; valuation

## 1. Introduction

Evidence has shown that extreme weather events, such as hurricanes and droughts, have increased in recent decades, and people are experiencing more frequent heat waves, heavy storms, and severe floods [1]. Climate variability and extreme weather events can have wide ranging impacts on agriculture including, but not limited to, crop stress due to high temperatures and impacts on crop yields due to high rainfall or drought [2]. Climate variability and extreme weather events can have direct impacts on people's well-being and property values. Secondary impacts such as costs associated with providing fresh water and many resources may be incurred days or weeks after the event [3]. In this context, the availability of and access to climate and meteorological information can help people respond in time to evacuate areas and thus save lives [4]. In agriculture, the availability of and access to climatic information such as seasonal climate forecasts, which provide probabilistic outlooks for a month to a year ahead, has significant potential to support and inform agricultural decisions [2]. Private providers lack sufficient incentives to deliver comparably timely weather and climate information because of the public nature of such information [5]. Since the information is available to users at no additional cost, governments, who are most often the providers of such information, may not know the potential benefit of the taxpayers' money. Finding a way to assign a value to the information, in the absence of a price, would help decision makers conduct cost–benefit analyses to decide whether to continue to allocate resources to providing weather and climate information [5].

Since climatic and meteorological information is widely used to support the decision making of local, regional, tribal, state and national agencies, efforts have been made to value the information. The valuation of climatic and meteorological information can be used to estimate the societal benefits from government investment and understand the societal benefits of disaster prevention and management. Research has been conducted in both developed and developing countries, including Vietnam, China, Australia, Korea, Switzerland, and the United States [6–11]. Although many efforts have been made, efforts in some sectors have been more focused than others. For example, many studies have focused on farmers and local decision makers utilizing climatic and meteorological information. However, how meteorological information has been used in the health and land management sectors has been less examined. Additionally, a better understanding of the literature on the valuation of climatic and meteorological information can provide some useful methods and tools to guide future research on those less examined sectors.

Compared to the wide use of climatic information and services such as drought information in agriculture, drought has a broader impact on land management. Specifically, drought affects recreation and tourism, grazing, forest and timber, and has important indirect effects for the ecosystems and species that rely on water [12–14]. First, drought affects recreation and tourism. The public lands managed by the Bureau of Land Management offer more recreational opportunities than lands managed by any other federal agency, with 99% of BLM-managed lands available for fee-free recreation [15]. Climate indirectly affects nature-based tourism by impacting the physical resources that define the nature and quality of natural environments on which mountain tourism depends (i.e., climate-induced biophysical change). Although access to public lands is free, there is an economic benefit to local economies surrounding lands used for recreational activities. Drought is also associated with public health issues such as water-related diseases, airborne and dust-related diseases, vector-borne diseases, mental health, and increased potential for wildfire-related illness and infectious diseases.

The first part of this paper discusses methods that have been used for assigning a monetary value to climatic and meteorological information. Methods and tools that have been explored to examine the benefits of climatic and meteorological information include the avoided cost, contingent valuation, choice experiments, benefit transfer and some descriptive approaches using surveys to understand the use of this information. The second part of this paper discusses the impact of drought in land management and public health. The majority of the papers that we summarized only report biophysical change, because the economic losses associated with drought are not available. Only a few papers reported economic losses associated with drought. The economic losses are not always the economic value of the information. We do not know exactly what role drought information plays in decision making and what specific sources of drought information are used in different sectors. Through the exercise, we hope to better understand the impact of drought in land management and public health. Additionally, we hope that this study can provide insight for future research on the value of drought information in the land management and public health sectors.

## 2. Literature Review

This section summarizes and systemizes a range of previous efforts, from the valuation of public information on drought, climate, and early warning systems to other metrological products such as weather forecasts. Existing literature tends to fall into two categories: applied economic studies, and literature reviews. Methods and tools that have been explored to examine the benefits of climatic and meteorological information include the avoided cost, contingent valuation, choice experiments, benefit transfer, and descriptive approaches using surveys to understand the use of this information. The keywords we used for our literature search were "valuation", "cost–benefit", "drought information", "climate information", and "meteorology information". We conducted the literature search through Web of Science and Google Scholar. We found 16 peer-reviewed papers and two reports. Different types of information have been valued in the literature, including an early warning system for tropical cyclones, a heat watch/warning system, forecasted drought information,

meteorological information, and weather (forecast) information worldwide (Table 1). We found that that there have been significant efforts to understand the value of the information. Mutiple studies summarized the literature, including case studies, and methods of valuation of hydro-meteorological systems and meteorological information [16–20]. In addition, there was a number of empirical studies focused on different sectors (agriculture, residential, energy, health, multiple sectors or all sectors). We reviewed the methods that have been used in the valuation of climatic and meteorological information. We found four basic types of methods: direct and indirect valuation methods, and market and non-market approaches. For example, for information traded on the market, the value can be easily calculated based on the price reflected in people's purchase behaviors. However, for public information, it is important to note that there is no standard approach, and the valuation of information may vary by sector. For such information with non-market values, both contingent valuation and choice experiments can be used to examine the value of the information.

## 2.1. The Direct Market Approach

The direct market approach uses market prices and quantities to estimate the extractive direct use of the information or services sold and bought in the markets [28]. Internet and technology advances, particularly in mobile devices, enable information access at a very low cost compared to traditional media such as radio, TV, newspapers and magazines. With easy access to the internet via mobile devices, people can pay for weather or meteorological services that meet their needs. For example, in addition to weather forecast and visualization such as the radar that free weather apps offer, a paid weather app provides additional services and customization, such as an alarm if the heat index is forecast to hit 37.8 degrees Celsius (100 degrees Fahrenheit) [29]. This can help decision makers and vulnerable individuals prepare for extreme events and, if necessary, evacuate ahead of time. Nygård [30] estimated the costs of the Finnish marine monitoring program and used cost–benefit analysis to assess the value of environmental monitoring.

**Table 1.** Summary of the papers/reports examining the value of climatic and meteorological information.

| Authors | Types of Information | Sector | Study Area | Methods |
|---|---|---|---|---|
| Bernknopf et al. [21] | Remotely sensed information: the GRACE Drought Severity Index | Agriculture | All U.S. counties with corn production information | Bayesian decision framework |
| Sharda and Srivastava [22] | El Niño Southern Oscillation (ENSO) forecasted drought information | Residential | Auburn, AL (with 55,000 people) | The avoided cost |
| Nguyen et al. [6] | Early warning system for tropical cyclones | Residential | Vietnam | Choice experiments |
| Malik et al. [16] | Hydrometeorological systems | Multiple users/sectors | Worldwide | Literature review |
| Leviäkangas [17] | Meteorological information | N/A | | Literature review |
| Frei [10] | Meteorology and climatology information | Household, Ag, and energy | Switzerland | A case study using benefit transfer |
| Anaman and Lellyett [8] | Weather information | Household | Sydney, Australia | Contingent valuation |
| Rollins and Shaykewich [23] | Weather forecast automated telephone answering device | Multiple sectors (institutes, hotel/catering, construction, landscape/snow, recreation/sport, TV/film, agriculture, and others) | Toronto, Canada | Contingent valuation |
| Ebi et al. [24] | Heat watch/warning system | Health | Philadelphia, PA | The avoided cost |
| Freebairn and Zillman [18] | Meteorological services | N/A | Worldwide | Literature review |
| Park et al. [9] | Meteorological services | Household | Korea | Contingent valuation |
| Kenkel and Norris [11] | Real-time mesoscale weather information | Agriculture | Oklahoma | Contingent valuation |
| Frei et al. [25] | Meteorological services | Transportation | Switzerland | Surveys |
| Perrels et al. [19] | Weather and climate services | N/A | Europe | Literature review |
| Von Gruenigen et al. [26] | Meteorological services | Transportation/aviation | Switzerland | The avoided cost |
| Rogers and Tsirkunov [20] | Early warning system | N/A | Worldwide | Literature review |
| Lazo et al. [5] | Weather forecasts | Household | U.S. | Contingent valuation |
| Kite-Powell [27] | Physical Oceanographic Real-Time System | Navigation | Houston/Galveston, U.S. | The avoided cost |

### 2.2. The Indirect Market Approach

Market prices do not exist most of time because of the public nature of climatic and meteorological information. Another commonly used indirect method is the avoided cost. This method refers to the costs that people avoid by using climatic and meteorological information, such as estimating market expenditures that would have been incurred to prevent or mitigate the impacts of drought. Some disasters and damages can be avoided or reduced in agriculture, transportation, health, power generation, and recreation, and the benefits of these prevention and mitigation actions can be translated into monetary terms. For example, Ebi et al. [24] performed a cost–benefit analysis of the Philadelphia Hot Weather-Health Watch/Warning System (PWWS) and found that issuing an individual warning lowered daily mortality by approximately 2.6 lives on average. They chose to use a $4 million value of life in the benefit estimate. They conclude that the net benefit of the issued heat wave warning was approximately $468 million in the period 1995–1998. Kite-Powell [27] examined the National Oceanic and Atmospheric Administration PORTS information in navigation of Houston/Galveston and found that $11.9 million in direct annual benefits can be attributed to PORTS data with a reasonable degree of confidence. Von Gruenigen et al. [26] examined the economic benefit of meteorological services to Switzerland's domestic airlines by analyzing the use of terminal aerodrome forecasts at Zurich Airport. By extrapolating the results based on the number of flights, the total economic benefits of terminal aerodrome forecasts to Switzerland's domestic airlines add up to 13–21 million Swiss francs per year. Sharda and Srivastava [22] used scenario analysis to estimate the avoided cost (both in volumetric and economic savings) if water management adjustments are made based on El Niño Southern Oscillation (ENSO) drought forecast information. They estimated that the forecast information could have saved $1,183,308 and $491,783 during the winter to summer seasons in the periods 1999–2000 and 2007–2008, respectively.

### 2.3. The Stated Preference Method

The stated preference method has been widely used for estimating the economic value of public goods or services. It has also been used in the valuation of climatic and meteorological information, using both contingent valuation [5,8,9,11,23] and choice experiments [6]. The contingent valuation method attempts to measure the value that people place on a particular public good or service taken as a specific bundle of attributes [31]. The choice experiments method builds on the idea that the value of a good is a reflection of its characteristics, and it requires the respondents to compare and select (rank or rate) alternative combinations of goods and policy characteristics. Contingent valuation generally poses a written or verbal description of the change to be valued, while choice experiments pose the changes in terms of changes in the attributes of the item to be valued [31]. Contingent valuation typically asks respondents to state their value directly or indicate a range in which the value resides in an open-ended or closed-ended question, while survey respondents would be given alternatives to consider and asked to choose the preferred alternatives or rank the alternatives in choice experiments [31]. The contingent valuation method is often criticized due to protest bids. Several types of protest bids may occur: some survey respondents may not answer, some may give positive but invalid bids (outliers), and others may state zero value for a good or service that they actually value [32]. These bids may potentially bias the willingness to pay (WTP) results, and thus special attention and treatments are needed.

Kenkel and Norris [11] investigated real-time mesoscale weather information using the contingent valuation method among agricultural producers in Oklahoma. Their findings show that agricultural producers in Oklahoma do not appear to be willing to pay significant fees to access improved weather information, despite the perceived usefulness of weather information and the impact of weather on farm income and profitability. Results show that, on average, producers are willing to pay $5.83 per month for raw mesoscale data and $6.55 per month for the raw data plus value-added weather-related products. The anticipated income from user fees could cover as much as half of those costs using the conservative estimates of aggregate WTP for the raw weather data. Rollins and Shaykewich [23] used the contingent valuation method to examine the weather forecast automated telephone answering

device in multiple sectors (institutes, hotel/catering, construction, landscape/snow, recreation/sport, TV/film, agriculture, and others). They found the average value per call varied by commercial sectors, from $2.17 for agricultural users to $0.60 per call for institutional users, with an overall mean of $1.2 per call. This would result in an estimate of benefits to commercial users of $16,500,000 per year based on approximately 13,750,000 commercial calls annually. Park et al. [9] conducted a contingent valuation survey of 1000 randomly selected households in Korea in 2014 and found the economic value of metrological service nationwide is $445 million per year. Lazo et al. [5] focused on people's attitudes and behaviors about specific weather forecast information. The paper examined the sources, perceptions, uses and value of weather forecast information using the contingent valuation method. The results showed a net benefit of $26.4 billion a year ($31.5 billion in benefits minus $5.1 billion in costs) to households.

The use of choice experiments is relatively new for the non-market valuation of climatic and meteorological information, and very few choice-experiment studies have been undertaken. Nguyen et al. [6] is the only study that we found using a choice experiment to examine climatic and meteorological information. They estimated the benefits of an improved cyclone warning service to households in Vietnam. They performed choice-experiment surveys and found that the benefit estimates of the maximal improvements in a number of attributes of cyclone warning services (i.e., forecasting accuracy, frequency of update, and mobile phone-based warnings) are approximately $7.1–$8.1 per household, which would be an upper-bound estimate.

*2.4. Benefit Transfer*

The benefit transfer method, which is always referred to as the second-best approach, is often criticized because people's willingness to pay for the particular climatic and meteorological information may vary across sites and even time [33]. Errors may occur if researchers rely on prior studies and transfer others' estimates directly into their own analyses. Despite this caveat, the benefit transfer method serves as a low-cost screening technique for further valuation studies [33]. Frei [10] used the benefit transfer method to value the information of meteorology and climatology. In this pilot study, the author calculated the economic benefit of meteorological services in Switzerland by extrapolating the value of meteorological services in the household, agriculture, and energy sectors from previous literature worldwide. The study shows that it is hardly possible to estimate one single figure representing the total benefit of weather services in Switzerland. However, it is possible to determine specific answers concerning particular benefits within specific sectors.

## 3. Drought and Land Management

Drought information has been applied in an agricultural context in many regions of the world, notably Africa, Brazil, the U.S. and Australia [2,34]. For example, the U.S. Drought Monitor (USDM) is a weekly map of drought conditions across the United States, showing which parts of the United States are experiencing various degrees of short- and long-term drought. The U.S. Farm Bill has used the USDM as a trigger for various farm and ranch programs in the U.S. to the tune of over $7 billion in relief since 2012 alone [34]. Land provides the principal basis for human livelihoods and well-being including the supply of food, freshwater and multiple other ecosystem services, as well as biodiversity [35]. Land also plays an important role in the climate system [35]. Drought affects land management and ecosystem services, including the supply and quality of water resources for municipal, industrial and agricultural use [13–16]. Drought can also have important indirect effects for the ecosystems and species that rely on water. Thus, drought information can be useful for policy makers when making decisions to close down recreation sites, allocate land, and other resources such as issuing grazing permits, while protecting ecosystem services. Despite the importance, the use of drought information in land management has been less focused. In this section, we only focus on biophysical or economic losses associated with drought in land management due to the limitation of the literature.

### 3.1. Recreation and Tourisum Industry

The public lands managed by the Bureau of Land Management (BLM) offer more recreational opportunities than lands managed by any other federal agency, with more than 99% available for recreation with no fee—a public good, provided without a market. In 2016, BLM lands received more than 64.6 million recreation-related visits [36]. Climate indirectly affects nature-based tourism by impacting the physical resources that define the nature and quality of natural environments on which mountain tourism depends. Any changes in the natural characteristics of mountain environments could negatively influence tourism by reducing the perceived attractiveness of the region's mountain parks [37,38]. Although access to public lands is free, there is an economic benefit to local economies surrounding lands used for recreational activities. Two studies used the travel cost method to examine the value of recreation impacted by drought. Ward et al. [39] examined the marginal value of water for recreation at US Army Corps of Engineers reservoirs in Sacramento, California. Using the travel cost method, they found that annual recreational value per cubic meter of water varied from $0.005 to $0.5. Grossmann [40] also used the travel cost method to examine the value of water availability in a wetland recreation site in Europe. The results indicate that the economic values associated with maintaining minimum in-stream flows for boating pay off the public investment of augmentation of low flows. Additionally, the economy of the tourism industry can be influenced by drought. Drought conditions during the summer of 1988 contributed to widespread forest fires in Yellowstone National Park, which resulted in evacuations of campgrounds and seasonal visitor accommodations being closed four weeks earlier than normal [41]. Total annual visits to Yellowstone in 1988 were reduced 15 percent (compared to 1987) and park officials estimated that the forest fires resulted in a loss of tourism-related economic benefits of $60 million [41]. Schneckenburger and Aukerman [42] analyzed the economic effects of the 2002 drought on Colorado's recreation and tourism industry. The estimated revenue decline due to the 2002 drought was approximately 20 percent, i.e., $1.7 billion, both directly and indirectly in Colorado's tourism and recreation sectors. Enormous and severe drought impacts were found for state and county parks, the boating industry, the rafting industry, the fishing industry, the ski industry, and regional small businesses such as hotels and motels. Thomas et al. [43] found that 2012 drought impacts were not just economic, but extended to the very livelihood of communities and social well-being in Colorado. In another study, Leones et al. [44] examined the impacts of streamflow depletion on rafting businesses in northern New Mexico counties. They found that the lower water levels generally had negative effects on daily visitor numbers and rafting-related expenditures, but the magnitude of the impacts depended on the characteristics of the river courses.

### 3.2. Grazing

In 2016, the BLM permitted 12 million animal unit months (AUMs) for ranchers who graze their livestock—mostly cattle and sheep—on public lands. An AUM is the amount of forage needed to feed a cow and calf or the equivalent for one month [17]. In 2016, the grazing fee was $2.11 per AUM [36]. While the number of AUMs sold each year remains relatively steady, annual variations in use occur because of factors such as drought, wildfire, market conditions, and restoration projects [17]. Figure 1 shows the total number of grazing permits and leases from 2001 to 2017. The severe droughts in 2002 and 2012 may have influenced the total number of grazing permits and leases. Despite the fact that BLM and the Forest Service spend more managing the grazing programs than they collect in grazing fees, drought and wildfires may have an impact on the acreage available for grazing [37]. The droughts in 2002 and 2012 have contributed to $190,006 and $516,989 losses in terms of grazing fees correspondingly when compared with the number of AUMs in 2001, based on grazing fees of $1.43 per AUM in 2002 and $1.35 per AUM in 2012 [45]. However, we do not know to what extent drought was the determining factor. Additional analysis can focus on what factors play a role in decisions to reduce grazing permits and leases in drought years and on the marginal impact of drought information on grazing permits.

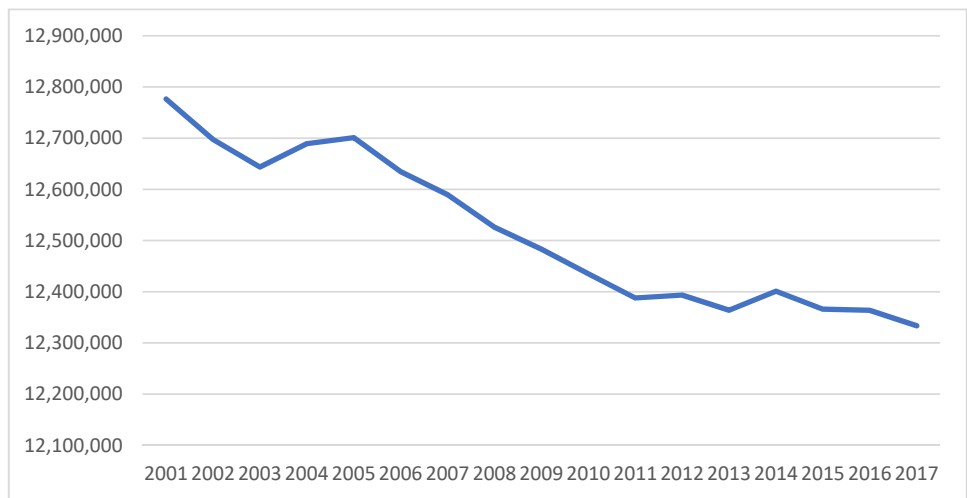

**Figure 1.** The total number of grazing permits and leases: animal unit months (AUMs) from 2001 to 2017. Source: BLM Public Land Statistics (BLM [46]).

### 3.3. Forest and Timber

Forest ecosystems cover approximately 30 percent of the Earth's land surface and provide numerous ecological, economic and aesthetic benefits across many spatial scales [47]. One-quarter of the 245 million acres of lands managed by the BLM are forest ecosystems, spread across 13 western states, including Alaska [17]. Through responsible management of these lands, the BLM ensures the health and resilience of the nation's public forest lands as well as the availability of traditional forest products, such as timber [17]. In 2016, the BLM offered 233.2 million board feet of timber for sale. This number has remained relatively steady over the past decade [26]. Forests are very vulnerable to drought and temperature extremes. Anderegg et al. [47] reviewed more than 150 studies documenting the global trend of forest mortality and identified 41 studies around the world on drought- and heat-induced forest mortality. Anderegg et al. [47] also found that extreme drought events can directly alter fungal, microbial and animal communities, which have an indirect impact on forest mortality. Drought may have an impact on timber production. Klos et al. [48] found that some tree species are sensitive to drought, while others are tolerant of drought. The observed differential growth and mortality rates among species groups may alter the species composition of southeastern U.S. forests if drought episodes become more frequent and/or intense. Drought-related forest mortality may further influence other ecosystem services that rely on forests. In some cases, forest mortality can increase the overall yield of the watershed and increase flood risks because of tree canopy loss. Forest mortality could also lead to increased runoff, water turbidity, erosion and stream siltation in steep terrain [47]. Forest die-off may increase the risk of wildfires and respiratory health-related issues. Forest mortality can also decrease cultural and esthetic values. For example, economics studies show that significant decreases in property values followed forest mortality in California and New England [47]. Price et al. [49] studied the impact of forest die-off on property values in Colorado. They found that property values declined by $648, $43, and $17 for every tree killed within a 0.1, 0.5, and 1.0 km buffer, respectively.

### 3.4. Other Ecosystem Services

Droughts can diminish water flow, thereby concentrating organisms and chemicals, and may reduce water for basic hygiene and decrease water quality in pools and lakes [50,51]. Other ecosystem services can also be greatly influenced by drought. We adapted the table of Compton et al. [52] and summarized ecosystem services and human benefits affected by drought in Table 2.

**Table 2.** Ecosystem services and human benefits affected by drought (adapted from Compton et al. [49]).

| Ecosystem Services | Drought Impacts on Human Benefits | Mechanism of Impact |
|---|---|---|
| Production of food and materials | − | Reduced production and nutritional quality of food crops |
| | +/− | Increased livestock production in the short term due to the water shortage; reduced livestock production in the long run during drought |
| | − | Reduced production of building materials and fiber for clothing or paper |
| | − | Reduced agricultural and wood production due to the lack of water |
| | − | Soil erosion, nutrient imbalances, altered species composition and diversity and other natural ecosystems, which ultimately impact stability and resistance to disease, invasive species and wildfires |
| Energy production | − | Reduced hydro-related power production; power plants may need to be shut down; a switch from coal to natural gas for utility providers; increased demand for air conditioning during heat wave/drought |
| | − | Increased demand for energy for pumping groundwater or surface water during drought |
| Clean air | − | Increased poor air quality due to dust or particles, dust bowl and wildfires |
| | − | Increased water demand from consumers |
| Drinking water | − | Increased airborne and dust-related disease such as Valley fever |
| | − | Increased contaminate levels in source waters can lead to diminished source water and can affect treatment costs and the ability to meet drinking water standards |
| | − | Increased outbreaks of waterborne diseases including Escherichia coli (*E.coli*) and other viruses |
| Swimming | − | Increased outbreaks of waterborne diseases including *E. coli* and leptospirosis |
| | − | Harmful algal blooms that make the water unsafe for swimming |
| Fishing | − | Increased hypoxia and harmful algal blooms in coastal zones, closing fish and shellfish harvests |
| | − | Increased hypoxia and harmful algal blooms in lakes and ponds |
| | − | Reduced number and species of recreational fisheries |
| Hiking | − | Altered biodiversity, health and stability of natural ecosystems |
| Aesthetics | − | Reduced scenery aesthetics due to altered biodiversity, unhealthy and unbalanced natural ecosystems |
| Visibility | − | Increased dust in air stimulates formation of particulates, smog and regional haze |
| Cultural and spiritual values | − | Altered biodiversity, food webs, habitat and species composition of natural ecosystems |
| | −<br>− | Damage to buildings and structures due to drought; people may abandon their houses and relocate<br>Greater impact on poor communities compared to more prosperous communities |

In sum, many researchers have documented the effects of drought on tourism and recreation. Closing recreation sites, reducing grazing permits and leases in drought years and altering the amount of timber form one potential BLM response to drought, and in-depth research could reveal the cost to local economies. However, the question of whether drought information was used in the decision—as opposed to an on-site assessment of local conditions—is a separate one, as would be the value of those tools in making the decision. For example, closing a recreation site would probably be a negative for the local economy, balanced against a public good such as protecting the long-term capacity of the site to provide ecosystem services.

## 4. Drought and Public Health

Severe drought conditions have been associated with widespread crop failure and food shortage, resulting in famine in developing countries [53]. In the United States, malnutrition and starvation have not been public concerns because of an advanced capacity for food production and distribution [53]. However, drought has been associated with crop failure and economic losses such as the higher cost of food and safe water [54]. Drought is also associated with public health issues such as water-related diseases, airborne and dust-related diseases, vector-borne diseases, mental health, and increased potential for wildfire-related illness and infectious diseases. This section focuses on biophysical or economic losses associated with drought in public health sector. Please note that the figures reported are economic losses associated with drought. For example, they can be hospital charges instead of the value of the drought information.

### 4.1. Effects on Nutrition

Drought causes more than 80 percent of the total damage and losses in agriculture, especially for the livestock and crop production subsectors [55]. If a drought is severe and widespread enough, it can potentially affect national food availability and access, as well as nutrition, thus magnifying the prevalence of undernourishment nationally [55]. In the United States, famine and undernourishment occurred when little or no drought relief assistance was provided by state and federal government [56]. It is very rare in the United States now. Most of the famine and undernourishment cases occur in countries where people's livelihood is highly dependent on agricultural production and their agricultural systems are also highly sensitive to rainfall, temperature variability and severe drought [55].

### 4.2. Water-Related Diseases

In the United States, more than 200 million people have direct access to disinfected public water supply systems, yet as many as 9 million cases of waterborne disease are estimated to occur each year [50]. As many as 900,000 cases and 900 deaths from waterborne disease occur annually in the United States [50]. These outbreaks often happen when contamination is followed by a breakdown in waste management or drinking water treatment [50]. Weather conditions such as wind, rain and drought can affect water quality, and high air temperatures can cause overcrowding and decreased water quality in pools and lakes [51]. Dramatic outbreaks affecting a community's water supply have been linked both to droughts and flooding [50]. Heavy rains following a period of drought coincided with a major outbreak event in the state of New York [50]. Cann et al. [57] conducted a systematic review to examine waterborne outbreaks following extreme water-related events such as excessive precipitation, floods, and droughts. In the largest acute gastrointestinal illness of unknown etiology, an estimated 9847 persons in Puerto Rico became ill when the water system resumed operation after an interruption due to drought [58]. Wastewater discharges into the river water source, temporary lack of chlorination, filtration deficiencies, and insufficient flushing of old water from pipes and tanks were identified as possible factors contributing to the outbreak [58]. Another study found that an outbreak of leptospirosis was associated with swimming in a rural pond that was stagnant and that drought conditions facilitated transmission of organisms from area animals to humans [59].

### 4.3. Airborne and Dust-Related Diseases

During a drought, soils become increasingly dry and dust circulated in the air is more likely [60]. Due to the small size of the dust particles, particles of approximately ~10 μm or less can penetrate the deepest part of the lungs and cause health problems [61]. Ultra-fine particles, which are smaller than 2.5 μm, may pass through the lungs to affect other organs, with possible cardiovascular consequences [61]. The United States dust bowl drought of the 1930s is a particularly well-known example of this: hundreds and perhaps thousands of people who lived in the Great Plains died from "dust pneumonia," a respiratory condition brought on by inhalation of excessive amounts of dust and dirt [60]. Smoke from drought-related fire has also been found to increase respiratory stress. Greenough et al. [50] found that drought-generated wildfires have less impact on mortality but cause an increased incidence of functional limitations and respiratory symptoms.

A soil-dwelling fungus and its arthroconidia (particles approximately 2–5 μm in length) cause Valley fever when inhaled [62]. Between 1990 and 2008, an average of 161 deaths were attributed to Valley fever every year in the United States [63]. In 2004, approximately 3000 cases of Valley fever were reported in Arizona and 6000 cases were reported nationwide. Arizona Hospital Discharge Data show 1735 hospital visits for Valley fever in 2007, resulting in $86 million in hospital charges alone. In 2016, hospitalization charges for Arizona residents with a primary diagnosis of Valley fever totaled $55 million, with a median of $47,212 in total charges per hospitalization. Flaherman et al. [64] counted an average of 70 deaths from Valley fever in neighboring California each year between 1997 and 2008. Medical records for Kern County, California, attribute approximately $45 million in direct costs for hospitalization and outpatient care for Valley fever cases during the period 1991–1993 [65]. Sondermeyer et al. [66] used the California Patient Discharge Dataset, developed by the Office of Statewide Health Planning and Development, to review hospitalization data for the period 2000–2011. They found the median length of stay per hospitalization was 6 days and the median total hospital charge per patient was $55062 during the period 2000–2011. They also found that the total charge for all Valley fever-associated hospitalizations in California was $2.2 billion, and the average annual total was $186 million from 2000 to 2011. A recent study found that 3089 deaths due to Valley fever occurred in the U.S. from 1990 to 2008, which is an average of fewer than 200 deaths per year [60]. The number of Valley fever-associated deaths each year has been fairly stable since 1997 [63]. Drought began to develop across the southern tier of the U.S. during the winter of the period 2010–2011, and quickly intensified during the 2011 growing season [67]. Drought charged northward and intensified during the spring and summer of 2012, and nearly two-thirds (63.86 percent) of the contiguous U.S. was in drought, according to the U.S. Drought Monitor [67]. The number of reported Valley fever cases peaked during the period 2010–2012, with total cases of 16,793, 22,641, and 17,802, respectively, as shown in Table S1.

### 4.4. Vector-Borne Diseases

In the United States, below-average rainfall the previous year tended to increase West Nile Virus (WNV) transmission the following year [68]. Human infection has increased from 6–8 to 17 per 100,000 because of severe drought. Wang et al. [69] compared infection rates with annual rainfall levels in Mississippi and found an inverse relationship between precipitation levels the previous year and the relative risk of human WNV. The results also suggested that drought acts as a potential mechanism for increased risk of human WNV transmission. Shaman et al. [70] examined the relationship between hydrological variability and the incidence of WNV in Colorado from 2002 to 2007 and found that dry spring and summer conditions appear to increase the risk of human WNV infection. Staples et al. [71] estimated that the total cumulative costs of reported WNV hospitalized cases from 1999 to 2012 was $778 million (95 percent confidence interval: $673 million–$1.01 billion) based on over 37,000 WNV disease cases, including 1500 deaths reported to the Centers for Disease Control and Prevention. These estimates can be used in assessing the avoided cost and cost-effectiveness of interventions to prevent WNV disease [71]. Supplementary materials Figures S1–S2 show the number of WNV total cases and

deaths during the period 2002–2017. In the 2002 drought, more than 50% of coterminous United States experienced moderate to severe drought conditions, with record or near-record deficit throughout the western United States [72]. As a consequence of the severe drought conditions of 2002 and 2012 in the United States, WNV total cases and deaths spiked in 2002, 2003, and 2012.

Stanke et al. [60] reviewed a number of papers that examined the impact of drought on various vector-borne diseases. Other than the WNV, Stanke et al. [60] summarized the literature on other vector-borne diseases such as dengue, malaria, St. Louis encephalitis virus, Rift Valley fever virus, Japanese encephalitis, chikungunya, tick-borne disease, schistosomiasis, and Chagas disease and found that drought conditions may contribute to outbreaks of these diseases.

*4.5. Mental Health*

Drought can also have a significant negative impact on mental health. Australian researchers have found that droughts have contributed to farmers' or ranchers' emotional stress during severe drought [73–75]. In cases of prolonged or severe drought, people may migrate or move to urban areas in search of employment [53]. Drought can also lead to psychological and social effects including decreased quality of life, major changes in lifestyle, and increasing conflict over water resources [53,54]. Carroll et al. [76] estimated the cost of droughts by matching rainfall data with individual life satisfaction in Australia from 2001 to 2004. They found that the quarterly drought episode is equivalent to the loss of A$18,000 in household income. Their estimate shows the psychological costs of the drought (A$5.4 billion), potentially through increased insecurity, and were comparable to the economic costs (A$6.6 billion).

*4.6. Other Health Effects*

In the United States, urban sprawl has resulted in more people at risk from wildfire injuries [48]. For example, a 1991 grass wildfire fueled by dry vegetation from five consecutive years of drought conditions resulted in 25 fatalities and 241 fire-related hospital emergency visits in Alameda County, California [77]. The deadliest and most destructive wildfire in California history to date, the Camp Fire in 2018, resulted in 86 civilian fatalities and 3 firefighter injuries, and destroyed more than 61,917 hectares (153,000 acres) of land and 14,000 residences and other buildings [78]. The cause of the wildfire is still under investigation, but the lingering drought in the area was a crucial factor.

Other drought-related incidences include heat vulnerability among the socially isolated, children, the poor, and the elderly [79]. Drought also affects other populations, including dialysis patients, the elderly, pregnant and nursing women, infants, immunocompromised individuals, and persons with preexisting health conditions, such as hypertension and diabetes [79].

To sum up, we found a multitude of connections between drought and health in the literature, but not a wealth of obvious starting points for public health practitioners. In contrast to the developing world, nutrition is probably not a top drought-related issue in the United States. Well-developed, diversified national economies of developed countries generally buffer their populations from direct food shortages related to drought, in contrast to countries where a high proportion of the population depends on subsistence agriculture. However, drought is an environmental contributor to air-, water- and vector-borne disease and adds to stress that can compound mental health issues. The way in which drought contributes to reducing human well-being is multifaceted—in many instances, it is a background issue, and not obviously tractable. Determining which issues are tractable and what the leverage points are for intervention is an important next step.

## 5. Summary and Conclusions

Decision makers in many sectors and contexts benefit from meteorological information such as drought information: households, land management, crop and livestock planning, disaster preparation and relief, hydropower, fisheries management, and more. Drought information is typically provided as a public good, along with information such as weather forecasts. Valuing public goods is an inherent

challenge, because, by definition, they are services that the private sector lacks incentive—a market—to provide. In the private sector, market-based revenue provides a ready gauge for whether to continue an activity.

This paper reviews the four different approaches to understanding the value of meteorological information in the literature: the direct market approach, the indirect market approach (the avoided cost method), the stated preference method, and the benefit transfer method. In order to understand the other potential costs that can be saved by using meteorological information, we chose drought information as an example because it is less studied. We focused on the impact of droughts in both the Bureau of Land Management and the public health sector and explored the role of drought information in decision making. Although specific dollar amounts associated with the potential avoided cost in these two sectors do not exist, we have listed the effects of drought in these two sectors. For example, BLM lands received more than 64.6 million recreation-related visits [36]. Closing recreation sites because of drought can have a huge impact on the recreation and tourism industries and also further damage the local economy. However, if the environmental benefits were included, the benefits to ecosystem services and the environment may outweigh the temporary loss in profits generated in recreation and tourism in the long run. As another example, droughts in 2002 and 2012 contributed to $190,006 and $516,989 losses, respectively, in terms of grazing fees when compared with the number of AUMs in 2001, based on grazing fees of $1.43 per AUM in 2002 and $1.35 per AUM in 2012. However, we do not know to what extent drought is playing an important role. Additional analysis can focus on what factors play a role in decisions to reduce grazing permits and leases in drought years and what is the marginal impact on grazing permits of drought information. The potential costs on ecosystem services and the environment may outweigh the loss in grazing fees if they were considered. This paper briefly reviewed the methods on the non-market valuation of meteorological goods and services. Due to the lack of literature on the value of drought information in land management and public health, we only focus on the damages caused by drought. Future analysis can use non-market valuation tools to examine the value of drought information in these two sectors. For example, it would help to obtain better information on the number of health officials that are dealing with drought-related problems. Case studies of specific health offices could identify the primary public health threats associated with drought, and how timely information could be used to mitigate the effects.

This study identified gaps in available information, policy approaches, and challenges facing the valuation of meteorological information. This study explored the benefit transfer approach to value drought information in the land management and public health sectors. The effective use of the information can potentially help agencies and institutions avoid higher costs due to drought and can be further used as a proxy for benefits in a cost–benefit analysis. Our case studies examined the value of the impact of drought within the land management and public health sectors. Future direction can focus on surveying federal employees at BLM and public health practitioners on the willingness to pay for drought information and the avoided cost of using drought information. Subsequent research can build on the study and develop a more rigorous estimate of the value of information provided in specific contexts and compare the short-term and long-term benefits by using drought information. A more detailed understanding of the value of drought information can be helpful for policy makers in decision making related to investing in public goods and services in the face of droughts.

**Supplementary Materials:** The following are available online at http://www.mdpi.com/2073-4441/12/4/1064/s1, Figure S1: Number of West Nile Virus disease total cases (2002–2017). Source: CDC (2019a), Figure S2: Number of West Nile Virus Disease Deaths (2002–2017). Source: CDC (2019b), Table S1: Number of reported Valley fever cases.

**Author Contributions:** T.L. collected and complied the literature; T.L. and K.H.S. wrote the paper, R.K.; T.H.; and M.S. provided comments and suggestions on the paper. We thank Deborah Wood for her excellent professional editing. All authors have read and agreed to the published version of the manuscript.

**Funding:** This research was funded by the National Oceanic and Atmospheric Administration's Sectoral Applications Research Program and by the National Integrated Drought Information System, through award number NA19OAR4310402.

**Conflicts of Interest:** The authors declare no conflict of interest.

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
