# Peer review of "Critical Analysis of the Value of Drought Information and Impacts on Land Management and Public Health"

_water, doi:10.3390/w12041064_

Round 1
Reviewer 1 Report
The paper presents an interesting review of the efforts to assign monetary value to climatic or meteorological information, which is quite novel and original and is of undoubted interest to Water readers.
The following aspects of this work can be improved:
1) The need to explain in more detail the criteria of bibliographical review that lead to conclude that the areas of Public Health and Land Management have the greatest interest to focus the attention of this research paper and, in addition, have been less attended by the literature.
2) Also, the need to further specify the reasons that explain the focus on information on droughts and not on other risks. Shouldn't the word drought be mentioned in the title to be more clarifying about the content of the paper?
The approach and development of the paper is considered excellent, so it would only be recommended to review the issues identified above.
Author Response
Response to the reviewer #1:
We first want to thank Reviewer #1 for a careful reading of our manuscript. In the reviewer’s letter, there were two major comments. We have responded to both of them and incorporated the changes into our revised manuscript. We believe that because of the changes, the paper is much improved.
In order to facilitate the responses, we first paraphrase the reviewer’s comments. We then provide our response to each comment in the indented discussion that follows each one.
Comment 1: Why did the author choose to focus on land management and public health sector?
[Response to Comment 1] We thank the reviewer for the comments. We added the following justification to the main text.
On page 2: Compared to the wide use of climatic information and services such as drought information in agriculture, drought has a broader impact on land management. More specifically, it affects recreation and tourism, grazing, forest and timber, and has important indirect effects for the ecosystems and species that rely on water (Chang and Bonnette, 2016; Covich, 2009; Rosegrant, 1997). First, it affects recreation and tourism. The public lands managed by the Bureau of Land Management offer more recreational opportunities than lands managed by any other federal agency, with more than 99% available for recreation with no fee (BLM, 2017b). Climate indirectly affects nature-based tourism by impacting the physical resources that define the nature and quality of natural environments on which mountain tourism depends (i.e., climate-induced biophysical change). Although access to public lands is free, there is an economic benefit to local economies surrounding lands used for recreational activities. Drought is also associated with public health issues such as water-related diseases, airborne and dust-related diseases, vector-borne diseases, mental health, and increased potential for wildfire-related illness and infectious diseases.
Comment 2: The reviewer suggested to have “drought” in the title since because the manuscript summarized damages caused by drought in land management and public health.
[Response to Comment 2] We thank the reviewer for the comments. We changed the title to “Critical analysis of values of drought information and impacts on land management and public health”

Reviewer 2 Report
The paper summarizes the knowledge available in the US about the impacts that drought can have on land management and public health. It is specified in the conclusion that no economic knowledge is available on the impact that drought can have on land management and public health. However, examples of value of information (in particular weather forecast) are provided. These examples can be replicated to measure the economic benefit of knowing the value of monitoring to reduce damage to land management and public health.
I recommend the paper for publication with minor amendments as below reported
Introduction/Literature review
In the introduction, it is specified that the paper is considering the value of information (line 53). However, for valuation method like the direct market approach, what reported is not the value of information but the cost of accessing the weather forecast service. The value of information is the benefit derived for different economic sectors by choosing a different approach in land management (grazing, recreation, etc.) that limits damages having acquired information on drought. It would be better to specify at the end of the Introduction or in the Literature Review that not all the results proposed in section 2 (I refer in particular to the direct market approach) reflect this definition of value of information. The other methods reported in the literature review show correct example of the value of information.
For the meaning of value of information, but applied to another context, please refer to this paper.
Nygård, H., Oinonen, S., Hällfors, H.A., Lehtiniemi, M., Rantajärvi, E., Uusitalo, L. 2016. Price vs value of marine monitoring. Frontiers in Marine Science 3:205 doi: 10.3389/fmars.2016.00205.
Line 154-157: I would make an adjustment to the sentence reported at these lines because the economic benefit of information is given only by the additional WTP of $0.75 per household per month.
At the beginning of section 3 and 4, but probably better in the introduction when describing the structure of the paper, it is needed to report that papers reported in sections 3 and 4 refer to biophysical change in land use management and public health, but economic estimate are not available.
Section starting at line 198: it seems that the examples and facts reported refer to the economic impacts to tourism such as direct revenue lost, jobs etc. rather than the value of recreation. Do you have any specific example where the value of recreation (measured by travel cost method, for example) has been altered by drought? Can you be more precise in this section when you refer to recreation or economy of the tourism industry?
Section on grazing, from line 225: From figure 1, I see that there is a constant decline in grazing permits. You presume that this can also be imputed to drought. However, we do not know to what extent drought is playing an important role without estimating by regression analysis its marginal impact on the number of permits issued. You may provide more comments about this point here or in the conclusions.
Line 256: you state that forest mortality can decrease water turbidity. I was expecting the opposite, an increase in turbidity due to an excess of soil erosion and therefore a reduction in the quality of water.
Line 266: typo “…can also be greatly influenced by drought”
Line 315: referring to the section Airborne and dust-related diseases, please specify that the economic figures reported are the costs of accessing the hospitals, but this is different from the value of information.
Line 350: you mention “In the United States, below-average rainfall the previous year tended to increase WNV transmission the following year”. I presume that WNV is West Nile Virus. Have you mentioned this before? I have not found it (maybe I missed it). If not, state in full the meaning of WNV.
Line 433: you mention that this study explored the BT (Benefit Transfer) approach to value drought information. I think that you reported a review of cases of the damages caused by drought and how several methods in the applied economic literature can be used to value information on drought. It seems to me that the paper is not addressing the BT approach to your area of investigation.
Conclusions: please report clearly gaps in knowledge and methods and research needed to address the value of information to land management and public health affected by drought. Specify this better mainly for the readers who are not familiar with economic analysis of environmental goods.
Author Response
Response to the reviewer #2:
We first want to thank Reviewer #2 for a careful reading of our manuscript. In the reviewer’s letter, there were two major comments and several minor comments. We have responded to all of them and incorporated the changes into our revised manuscript. We believe that because of the changes that our paper is much improved.
In order to facilitate the responses, we first paraphrase the reviewer’s comments. We then provide our response to each comment in the indented discussion that follows each one.
Comment 1: In the introduction, it is specified that the paper is considering the value of information (line 53). However, for valuation methods like the direct market approach, what is reported is not the value of information, but the cost of accessing the weather forecast service. The value of information is the benefit derived for different economic sectors by choosing a different approach in land management (grazing, recreation, etc.) that limits damages having acquired information on drought. It would be better to specify at the end of the Introduction or in the Literature Review that not all the results proposed in section 2 (I refer in particular to the direct market approach) reflect this definition of value of information. The other methods reported in the literature review show correct examples of the value of information. At the beginning of section 3 and 4, but probably better in the introduction when describing the structure of the paper, it is needed to report that papers reported in sections 3 and 4 refer to biophysical change in land use management and public health, but economic estimates are not available.
[Response to Comment 1] We thank the reviewer for the comments. The majority of the papers that we summarized only report biophysical change, because the damage costs are not available. Only a few papers reported economic loss associated with drought. The economic losses are not the value of the drought information, but utilizing drought information may help potentially mitigate drought impacts in some degree. However, we do not know exactly how important the role of drought information plays in decision making and what information is used in different sectors. Through the exercise, we hope to better understand the impact of drought on land management and public health. Additionally, we hope this study can shed some light on future research on the value of drought information in the land management and public health sectors.
On page 2: The first part of this paper discusses methods that have been used for assigning a monetary value to climatic and meteorological information. Methods and tools that have been explored to examine benefits of climatic and meteorological information include avoided cost, contingent valuation, choice experiments, benefit transfer as well as some descriptive approaches using surveys to understand use of these information. The second part of this paper discusses the impact of drought in land management and public health. The majority of the papers that we summarized only report biophysical change, because the economic losses of drought are not available. Only a few papers reported economic loss associated with drought. The economic losses are not always the economic value of the information. We do not know exactly what role drought information plays in decision making and what specific sources of drought information are used in different sectors. Through the exercise, we hope to better understand the impact of drought in land management and public health. Additionally, we hope this study can provide insight to inform future research on the value of drought information in the land management and public health sectors.
On page 3: We added: “ Nygård [23] estimated the costs of the Finnish marine monitoring program and used cost-benefit analysis to assess the value of environmental monitoring.”
[References]:
Nygård, H., Oinonen, S., Hällfors, H.A., Lehtiniemi, M., Rantajärvi, E., Uusitalo, L. 2016. Price vs value of marine monitoring. Frontiers in Marine Science 3:205 doi: 10.3389/fmars.2016.00205.
Comment 2: Line 154-157: I would make an adjustment to the sentence reported at these lines because the economic benefit of information is given only by the additional WTP of $0.75 per household per month.
[Response to Comment 2] We revised the South Korea example.
On page 4: Park et al. [9] conducted a contingent valuation survey of 1,000 randomly selected households in Korea in 2014, and found the economic value of metrological service nationwide is $445 million per year.
Comment 3: Section starting at line 198: it seems that the examples and facts reported refer to the economic impacts to tourism such as direct revenue lost, jobs etc. rather than the value of recreation. Do you have any specific example where the value of recreation (measured by travel cost method, for example) has been altered by drought? Can you be more precise in this section when you refer to recreation or economy of the tourism industry?
[Response to Comment 3] We thank the reviewer for the comments. We did a literature search and added case studies examining the value of recreation impacted by drought. Two studies used the travel cost method to examine the value of recreation impacted by drought. Ward et al. [41] examined the marginal values of water for recreation at US Army Corps of Engineers reservoirs in Sacramento, California. Using the travel cost method, they found that annual recreational value per cubic meter of water varied from $0.005 to $0. Grossmann [] also used the travel cost method to examine the value of water availability in a wetland recreation site in Europe. The results indicate that the economic values associated with maintaining minimum in-stream flows for boating pay off the public investment in augmentation of low flows. We changed the subtitle to “recreation and tourism industry”.
On page 5: We added: “Two studies used the travel cost method to examine the value of recreation impacted by drought. Two studies used the travel cost method to examine the value of recreation impacted by drought. Ward et al. [41] examined the marginal values of water for recreation at US Army Corps of Engineers reservoirs in Sacramento, California. Using the travel cost method, they found that annual recreational value per cubic meter of water varied from $0.005 to $0.Grossmann [] also used the travel cost method to examine the value of water availability in a wetland recreation site in Europe. The results indicate that the economic values associated with maintaining minimum in-stream flows for boating pay off the public investment of augmentation of low flows. Additionally, the economy of the tourism industry can be influenced by drought.”
[References]:
Grossmann, M., 2011. Impacts of boating trip limitations on the recreational value of the Spreewald wetland: a pooled revealed/contingent behaviour application of the travel cost method. Journal of Environmental Planning and Management, 54(2), pp.211-226.
Comment 4: Section on grazing, from line 225: From Figure 1, I see that there is a constant decline in grazing permits. You presume that this can also be imputed to drought. However, we do not know to what extent drought is playing an important role without estimating by regression analysis its marginal impact on the number of permits issued. You may provide more comments about this point here or in the conclusions.
[Response to Comment 4] We thank the reviewer for the comments. We added comments as suggested.
On page 6: However, we do not know to what extent drought was the determining factor. Additional analysis can focus on what factors play a role in decisions to reduce grazing permits and leases in drought years and on the marginal impact of drought information on grazing permits.
Comment 5: Line 256: you state that forest mortality can decrease water turbidity. I was expecting the opposite, an increase in turbidity due to an excess of soil erosion and therefore a reduction in the quality of water.
[Response to Comment 5] Yes. We changed the sentence to “Forest mortality could also lead to increased runoff, water turbidity, erosion and stream siltation in steep terrain”. We also fixed the citation.
On page 7: We edited: “Forest mortality could also lead to increased runoff, water turbidity, erosion and stream siltation in steep terrain.”
Comment 6: Line 266: typo “…can also be greatly influenced by drought”
[Response to Comment 6] The typo is fixed.
Comment 7: Line 315: referring to the section Airborne and dust-related diseases, please specify that the economic figures reported are the costs of accessing the hospitals, but this is different from the value of information.
[Response to Comment 7] We thank the reviewer for the comments. We have stressed “Please note that the economic figures reported are damage costs associated with drought. In this case, they are hospital charges, not the value of the drought information”.
On page 8: We added: “This section focuses on biophysical or economic losses associated with drought in public health sector. Please note that the figures reported are economic losses associated with drought. For example, they can be hospital charges instead of the value of the drought information”.
Comment 8: Line 350: you mention “In the United States, below-average rainfall the previous year tended to increase WNV transmission the following year”. I presume that WNV is West Nile Virus. Have you mentioned this before? I have not found it (maybe I missed it). If not, state in full the meaning of WNV.
[Response to Comment 8] Yes, WNV is West Nile Virus. We fixed the error.
Comment 9: Line 433: you mention that this study explored the BT (Benefit Transfer) approach to value drought information. I think that you reported a review of cases of the damages caused by drought and how several methods in the applied economic literature can be used to value information on drought. It seems to me that the paper is not addressing the BT approach to your area of investigation.
[Response to Comment 9] We thank the reviewer for the comments. The paper briefly reviewed the methods on non-market valuation of meteorological goods and services. Due to the lack of literature on the value of drought information in land management and public health, we only focus on the damages caused by drought. Future analysis can use non-market valuation tools to examine the value of drought information in these two sectors. The main text has reflected changes.
On page 9:
The paper briefly reviewed the methods on non-market valuation of meteorological goods and services. Due to the lack of literature on the value of drought information in land management and public health, we only focus on the damages caused by drought. Future analysis can use non-market valuation tools to examine the value of drought information in these two sectors. For example, it would help to have better information on number of health officials that are dealing with drought-related problems. Case studies on of specific health offices could identify the primary public health threats associated with drought, and how timely information could be used to mitigate the effects.
